

# Expression of CK7, CK19 and p16 in HPV-mediated oropharyngeal squamous cell carcinoma

Qizhang Yan[1,2,3,4,5,6], Pengning Chen[4,5,6,7], Xinyu Chen[4,5,6,7], Guanxi Chen[4,5,6,7], Lin Luo[4,5,6,7], Ping Ruan[8], Dahai Yu[9], Xiaojuan Zeng[1,2,3] and Mengyu Zhou[9]

[1] Department of Dental Public Health, Guangxi Medical University College of Stomatology, Nanning, China
[2] Department of Oral Health Policy Research, Guangxi Medical University College of Stomatology, Nanning, China
[3] Guangxi Health Commission Key Laboratory of Prevention and Treatment for Oral Infectious Diseases, Nanning, China
[4] Guangxi Key Laboratory of Oral and Maxillofacial Rehabilitation and Reconstruction, Nanning, China
[5] Guangxi Clinical Research Center for Craniofacial Deformity, Nanning, China
[6] Key Laboratory of Research and Application of Stomatological Equipment, College of Stomatology, Hospital of Stomatology, Guangxi Medical University, Nanning, China
[7] Department of Oral and Maxillofacial Surgery, Guangxi Medical University College of Stomatology, Nanning, China
[8] Department of Pathology, Ruikang Hospital Affiliated to Guangxi University of Chinese Medicine, Nanning, China
[9] Department of Stomatology, The First Affiliated Hospital of Guangxi Medical University, Nanning, China

Corresponding authors
Xiaojuan Zeng,
xiaojuan.zeng@gxmu.edu.cn
Mengyu Zhou,
zhoumengyu@gxmu.edu.cn

## ABSTRACT

**Background:** The incidence of oropharyngeal squamous cell carcinoma (OPSCC) mediated by human papilloma virus (HPV) has been steadily increasing worldwide. The specific pathogenesis of HPV-mediated head and neck squamous cell carcinoma (HNSCC) usually induces carcinogenesis in the oropharynx and the roles of CK7, CK19 and p16 in the carcinogenesis mechanism of HPV-mediated OPSCC still remain uncertain.

**Methods:** We collected case data and paraffin samples of 69 cases of OPSCC and 40 cases of OSCC from July 2009 to December 2021. Immunohistochemistry was performed on serial paraffin sections from all cases to analyze the expression patterns of CK7, CK19, and p16. HPV-mediated (p16+) and non-HPV-mediated OPSCC were differentiated based on p16 expression. Three to six fields were selected from each case for observation of the expression intensity, localization, and interrelationship of the three proteins.

**Results:** In both cancerous nests and pericancerous normal epithelium of OPSCC, various expression combinations of CK7, CK19 and p16 were observed, including CK7+CK19+p16+, CK7−CK19+p16+, and CK7+CK19+p16−, while no expression of CK7+CK19−p16+ was found. The expression of CK7 with CK19, CK7 with p16, and CK19 with p16 all showed consistency in OPSCC ($P < 0.05$) while only the expression of CK7 and CK19 demonstrated consistency in OSCC ($P < 0.05$). The positive rates and H-scores for CK7/CK19 in HPV-mediated (p16+) OPSCC were significantly higher than those in non-HPV-mediated OPSCC and OSCC ($P < 0.05$).

**Conclusions:** These results suggest that CK7, CK19 and p16 may relate to HPV-mediated OPSCC. The interaction of CK7, CK19 and p16 may affect the development of HPV-mediated OPSCC.

## INTRODUCTION

Oropharyngeal squamous cell carcinoma (OPSCC) is a squamous differentiated carcinoma deriving from the oropharyngeal mucosal epithelium, arising from the soft palate, base of the tongue (the posterior third beyond the sulcus terminalis), lateral pharyngeal wall, posterior pharynx wall, and tonsil (*Amin, 2017*). The etiology of OPSCC is considered to be long-term smoking and excessive drinking as well as human papilloma virus (HPV). Since the 1970s, the incidence of head and neck squamous cell carcinoma (HNSCC) has decreased due to a decline in the smoking rate in European and American countries (*Krane, 2013*). Although the incidence of HNSCC has decreased in multiple anatomic sites of the head and neck, OPSCC has been increasing and currently accounts for 20–25% of all HNSCC cases (*Krane, 2013*). HPV is found in many anatomic sites of head and neck carcinomas but typically induces carcinogenesis in the oropharynx, particularly in the tonsil and base of the tongue. In the United States, HPV infection accounts for 45–90% of OPSCC cases (*Psyrri, Rampias & Vermorken, 2014*). Among these HPV-mediated OPSCCs, the high-risk subtype, HPV-16, is the most common (*Jordan et al., 2012*). Predisposing risk factors for HPV+OPSCC include having multisexual partners oral sex or intercourse and cannabis use rather than alcohol and tobacco abuse (*Huang et al., 2013*), so the increase in HPV+OPSCC may be related to the modern lifestyle change. However, the specific pathogenesis of HPV-mediated OPSCC remains uncertain.

Cytokeratin 7 (CK7) is a marker of monolayer and columnar epithelium and is expressed in almost all normal epithelial tissues, especially in esophago-gastric junction region and squamou-columnar junction (SCJ) region of the cervix and oropharynx, except in the epithelia of liver, intestinal and prostate (*Moll, Divo & Langbein, 2008*; *Morbini et al., 2015*; *Xu et al., 2018*; *Hashiguchi et al., 2019*). HPV E6/E7 mRNA was detected in CK7-positive SCJ cells of cervical cancer, indicating that SCJ cells play an important role in cervical carcinogenesis (*Mirkovic et al., 2015*). It has been reported that CK7 is associated with oropharyngeal carcinoma caused by HPV infection, and its carcinogenesis may originate from tonsillar crypt reticular epithelium rather than surface epithelium (*Woods et al., 2017*). Studies have found that the SCJ region of the cervix and anus which has high expression of CK7, are recognized as susceptible regions for HPV infection (*Mirkovic et al., 2015*; *Yang et al., 2015*). Tonsillar crypt reticular epithelium may also be a susceptible region for HPV due to CK7 expression. Cytokeratin 19 (CK19), the cytokeratin with the smallest molecular weight, is an acidic polypeptide that constitutes the cytoskeleton. Compared with CK7, the tissue distribution of CK19 is similar but more widespread. CK19 is expressed in most of monolayer epithelia, such as the catheter, intestinal and gastric

foveal epithelium, and mesothelial cells. Additionally, it is found in most cells of pseudostratified and urothelial epithelia, and in basal cells of nonkeratinized stratified squamous epithelium, especially co-expressed with CK7 in monolayer epithelium (*Barrett et al., 2005*; *Moll, Divo & Langbein, 2008*).

There are significant differences in cytokeratin expression between the oral cavity and oropharynx. Oral mucosa, like the dorsum of the tongue, hard palate and gingiva, exposed to the forces of mastication and friction, and lined by keratinizing stratifed squamous epithelium, do not express either CK7 or CK19, while other parts of the oral cavity and oropharynx which are lined by non-keratinizing stratifed squamous epithelium, do show basal layer expression of CK19, but not CK7 (*Lindberg & Rheinwald, 1989*; *Depondt et al., 1999*; *Chu & Weiss, 2002*). By contrast, specific expression of CK7 and CK19 has been demonstrated in tonsil crypt epithelium, raising the speculation that these cytokeratins may play a role in HPV-associated carcinoma pathogenesis, and the specificity of these tumors for tonsillar epithelium (*Woods et al., 2017*). CK7 and CK19 have been previously associated with oral, oropharyngeal, and cervical squamous cell carcinoma (SCC) (*Kanduc, 2002*; *Favia et al., 2004*; *Lucchese et al., 2010*; *Park et al., 2010*; *Santoro et al., 2015*; *Woods et al., 2017*) and oral precancerous lesions (*Lindberg & Rheinwald, 1989*; *Babiker et al., 2014*).

p16INK4A (p16), a tumor suppressor gene directly acting on cell cycle and inhibiting cell division, is overexpressed in many HPV-mediated cervical cancers as well as OPSCC (*Chen et al., 2012*; *Lechner et al., 2022*). The oncogenic activity of HPV in HPV-associated carcinoma is mainly related to the E7 expression in tumor cells. E7 binds to and promotes Retinoblastoma Protein (pRb) degradation, resulting deregulation of the cell cycle and thus leading to the abundant expression of the p16 protein (*Serra & Chetty, 2018*). It is currently recognized as a surrogate marker for HPV infection (*Smeets et al., 2007*; *Hafkamp et al., 2008*; *Krane, 2013*; *Serra & Chetty, 2018*).

Currently, the roles of CK7, CK19 and p16 in the carcinogenesis mechanism of HPV-mediated OPSCC, as well as the reciprocal space-time effects and expression patterns of these three proteins remain uncertain. Morever, the expression relationship of the three proteins in HPV-mediated OPSCC is still unknown, especially the comparison of their expression relationship in the same site, such as in the cancerous nest, pericancerous crypt epithelium and surface epithelium. In this study, the OPSCC cases, including tonsil, base of the tongue and soft palate, were selected. And OSCC cases were used as controls. CK7, CK19 and p16 were detected in the serial paraffin sections of OPSCC/OSCC by immunohistochemistry (IHC) in order to compare the expression relationship of the three proteins in the same site, and to explore the relationship between these proteins and the carcinogenesis mechanism of HPV-mediated OPSCC based on the comparison between HPV-mediated (p16+) and non-HPV-mediated OPSCC as well as OSCC.

## MATERIALS AND METHODS

### Samples

A total of 109 patients including 69 cases of OPSCC and 40 cases of OSCC, were recruited from The First Affiliated Hospital of Guangxi Medical University and Guangxi Medical

University College of Stomatology (Nanning, China) from July 2009 to December 2021. The criteria for patient inclusion were as follows: (1) patients were diagnosed as squamous cell carcinoma (SCC) by hematoxylin-eosin (HE) staining and postoperative pathology was consistent with the preoperative diagnosis; (2) the primary sites of OPSCC patients were tonsil, base of the tongue and soft palate and of OSCC patients were tongue, buccal and gingiva; there were no other primary and/or distant metastases cancers during the same period; (3) patients had not received preoperative radiotherapy, chemotherapy and immunosuppressant; (4) the case data were detailed and paraffin specimens were complete. Case data and pathological specimens were obtained from database and the pathology departments of both hospitals. All the pathological specimens were neutral-formalin-fixed and paraffin-embedded, and from each specimen, 10 serial paraffin sections at 5 μm were taken for HE and IHC staining.

## Ethics statement

Permission was obtained from the Ethics Committee of The First Affiliated Hospital of Guangxi Medical University (No. 2022021) and Guangxi Medical University College of Stomatology (No. 2022-KY-E- (149)). The ethical review board approved the consent procedure and execution of this project. This study was granted an exemption from obtaining consent.

## IHC

The primary antibodies of CK7 (Kit-0021), CK19 (Kit-0030) and p16 (MAB-0673), as well as secondary antibody (Kit-5010) were purchased from Maixin (Fujian, China). IHC staining for CK7, CK19 and p16 was performed on serial paraffin sections from the primary tumor of OPSCC/OSCC cases. After deparaffinization and rehydration, antigen retrieval was performed using Tris-EDTA (pH 8.0). Endogenous peroxidase activity was blocked with 3% hydrogen peroxide for 5–7 min. After washing in PBS, sections were incubated in primary antibodies overnight at 4 °C and in secondary antibody for 20 min at 37 °C. Color was developed with diaminobenzidine, and the sections were counterstained with hematoxylin. PBS, as a substitute for the primary antibody, was used as a negative control.

## IHC staining assessment

**p16** According to The 8th Edition AJCC Cancer Staging Manual (*Amin, 2017*) and Guideline For Pathological Diagnosis Of HPV Detection In Head And Neck Carcinomas From The College Of American pathologists (*Lewis et al., 2018*), p16 in non-keratinising SCC in which more than 70% of tumor cells show moderately or strongly positive staining in the nucleus and cytoplasm without HPV DNA/RNA detection is defined as HPV mediated (p16+) SCC. However, it has reported that the proportion of keratinising SCC in HPV+OPSCC patients' specimens in Asia is higher than that in other regions (*Toman et al., 2017*; *Meng et al., 2018*). Therefore, in this study, we defined keratinising or non-keratinising SCC in which more than 70% of tumor cells show moderately or strongly

positive staining in the nucleus and cytoplasm without HPV DNA/RNA detection as HPV mediated (p16+) SCC.

**CK7 & CK19** CK7 and CK19 IHC staining of tumor cells was scored using the H-score system by two pathologists at a multi-headed microscope by consensus. The intensity of staining present in tumor cells was given a Score from 0 to 3 (0 = none, 1 = mild, 2 = moderate, 3 = intense). The H-score was then derived from the cross-product of the intensity and the percentage of tumor cells staining (0–100%) (*Santoro et al., 2015*; *Woods et al., 2017*, *2022*).

## Statistical analysis

Statistical analysis was performed using SPSS 25.0 for Windows. The consistency of CK7, CK19 and p16 was analyzed by the Kappa consistency test. The chi-squared test was used on 2 × 2 contingency tables to analyze the differences in positive rates of CK7/CK19 among HPV-mediated (p16+) OPSCC, non-HPV-mediated OPSCC and OSCC groups. Because H-scores of CK7 and CK19 were not normally distributed, the Kruskal-Wallis test was used to analyze the differences between groups. $P$-values < 0.05 were considered statistically significant.

# RESULTS

## Demographic characteristics

The study group comprised 109 patients, including 69 patients with OPSCC and 40 patients with OSCC. Among the specimens of 69 OPSCC patients, 14 exhibited non-keratinizing morphology with p16+ tumor cells comprising more than 70%, and five exhibited keratinizing morphology with p16+ tumor cells comprising more than 70%, whereas the remaining 50 were either p16− or keratinising/non-keratinising morphology with p16+ tumor cells comprising less than 70%. Thus, the final HPV status was determined as 19 HPV mediated (p16+) OPSCC and 50 non-HPV-mediated OPSCC. Demographic and clinical details of patients with OPSCC according to HPV-status are given in Table 1.

## Expression of CK7/CK19/p16 in the cancerous nests and pericancerous normal epithelium of OPSCC/OSCC

Oropharyngeal mucosa, especially the tonsil and base of the tongue, is covered with stratified epithelium. Crypt epithelium, existing in the tonsil and the base of the tongue mucosa, consists of reticular epithelium. In contrary, oral mucosal epithelium is stratified squamous epithelium whitout crypt structure. In the oropharyngeal normal mucosal tissue (amygdalitis) control group, CK7, CK19 and p16 were positive in the normal tonsils (Figs. 1B–1D). However, CK7 and CK19 but not p16, were expressed in the oral mucosal epithelium (Figs. 2B–2D). In the pericancerous normal epithelium of OPSCC, CK7 was predominantly localized in the surface layer of the epithelium (Fig. 1F), while CK19 was primarily expressed in the subepithelial layer and basal cells (Fig. 1G). The expression of p16 was observed in the lower 1/3 to the entire layer of the epithelium (Fig. 1H), with

**Table 1 Clinical and demographic details of patients with OPSCC.**

| | HPV mediated (p16+) OPSCC | Non-HPV-mediated OPSCC | *P* value |
|---|---|---|---|
| **Sex** | | | |
| Male | 11 | 45 | 0.002 |
| Female | 8 | 5 | |
| **Age** | | | |
| <40 | 2 | 3 | 0.521 |
| 40–59 | 13 | 30 | |
| ≥60 | 4 | 17 | |
| **Smoking index** | | | |
| <20 pack years | 19 | 27 | <0.001 |
| ≥20 pack years | 0 | 23 | |
| **Nationality** | | | |
| The Han nationality | 11 | 33 | 0.855 |
| The Chuang nationality | 7 | 15 | |
| Another | 1 | 2 | |
| **Primary site** | | | |
| Tonsil | 11 | 21 | 0.326 |
| Base of tongue | 4 | 20 | |
| Soft palate | 4 | 9 | |
| **Degree of differentiation** | | | |
| Medium-low differentiation | 16 | 24 | 0.006 |
| Well-differentiated | 3 | 26 | |
| **Morphology** | | | |
| Keratinizing | 5 | 20 | 0.291 |
| Non-keratinizing | 14 | 30 | |
| **N stage** | | | |
| $N_0$ | 13 | 16 | 0.006 |
| N+ | 6 | 34 | |
| **T stage** | | | |
| 1–2 | 10 | 30 | 0.580 |
| 3–4 | 9 | 20 | |
| **Clinical stages** | | | |
| I–II | 17 | 11 | <0.001 |
| III–IV | 2 | 39 | |

significant overlap with the CK19 site. Conversely, the expression patterns of CK7, CK19 and p16 in the pericancerous normal epithelium of OSCC exhibit similarities but lack typical characteristics (Figs. 2F–2H). Additionally, our study has unveiled the presence of soft palatal lymphatic follicles and crypt epithelial structures that are positive for CK7, CK19 and p16 deep within the follicles, a novel finding in this context (Fig. 3).

The co-expression of CK7, CK19, and p16 in OPSCC/OSCC was presented in Table 2. The expression patterns of CK7, CK19, and p16 within the cancerous nests of HPV-mediated

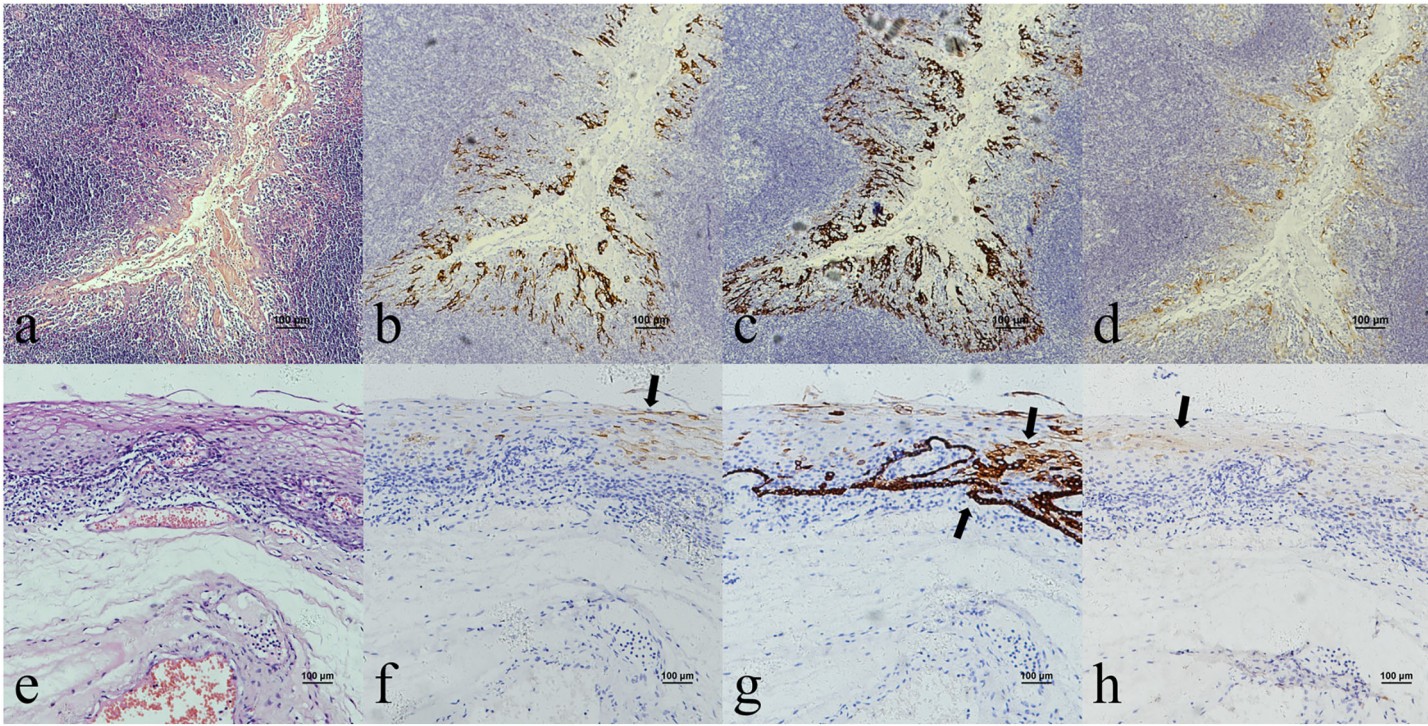

**Figure 1** **CK7, CK19 and p16 staining in the normal oropharyngeal (tonsilar) crypt epithelium and the pericancerous surface normal epithelium of oropharyngeal (tonsil) squamous cell carcinoma.** (A) HE staining of the normal tonsilar crypt epithelium. (B) CK7 showed patchy staining in the crypt epithelium. (C) CK19 showed patchy staining in the crypt epithelium. (D) p16 showed patchy staining in the crypt epithelium. (E) HE staining of OPSCC. (F) CK7 showed patchy staining in the epithelial surface layer (black arrow). (G) CK19 showed diffuse staining in the whole epithelial layer and in the basement membrane (black arrow). (H) p16 showed patchy staining in the lower 1/3 to the whole layer of the epithelium, which basically overlaps with the CK19 site (black arrow).

(p16+) OPSCC, non-HPV-mediated OPSCC, and OSCC were illustrated in Figs. 3–8. In addition to CK7+CK19+p16+ (Figs. 3–5), other combinations such as CK7−CK19+p16+ (Fig. 6) and CK7+CK19+p16− (Fig. 7) were also observed in both the cancerous nests and pericancerous normal epithelium of OPSCC/OSCC. Furthermore, individual positivity for CK7, CK19 or p16 was noted, with only one case exhibiting the expression of CK7+CK19 −p16−. It is noteworthy that in the cancerous nest and pericancerous normal epithelium of OPSCC/OSCC, only the expression situation of CK7+CK19−p16+ could not be observed (Table 2 and File S1).

## Comparison of CK7/CK19 expression between OPSCC and OSCC

The positive rates and H-score details of CK7 and CK19 in OPSCC/OSCC were given in Table 3.

The positive rates and H scores of HPV-mediated (p16+) OPSCC were substantially higher than those of non-HPV-mediated OPSCC and OSCC, as indicated by CK7. However, there was no significant difference in the positive rates and H scores between non-HPV-mediated OPSCC and OSCC.

The positive rates and H scores of HPV-mediated (p16+) OPSCC were significantly greater than those of non-HPV-mediated OPSCC, specifically for CK19. Moreover, the

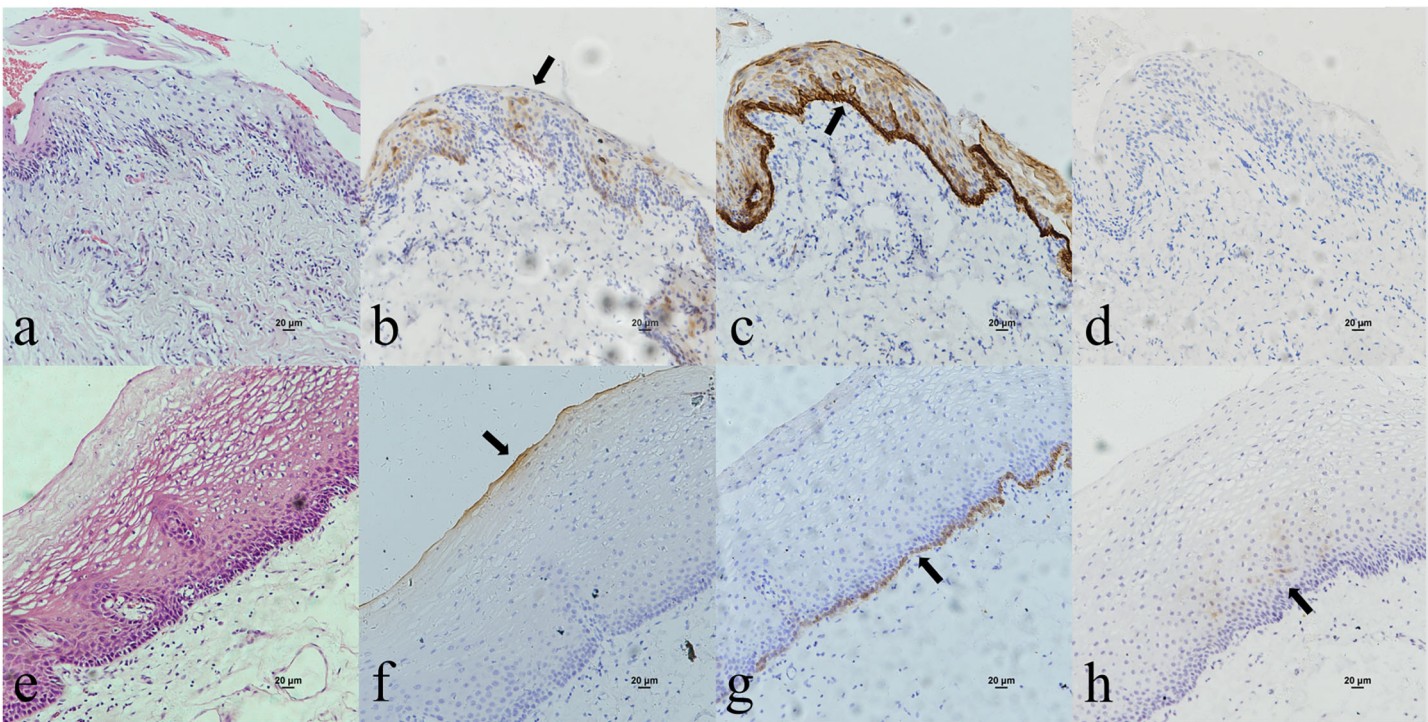

**Figure 2 CK7, CK19 and p16 staining in the oral mucosal normal epithelium and the pericancerous surface normal epithelium of oral (tongue) squamous cell carcinoma.** (A) HE staining of the oral mucosal normal epithelium. (B) CK7 showed patchy staining in the epithelial surface layer (black arrow). (C) CK19 showed diffuse staining in the whole epithelial layer and in the basement membrane (black arrow). (D) p16 was negative in the epithelium. (E) HE staining of the pericancerous surface normal epithelium of oral (tongue) squamous cell carcinoma. (F) CK7 showed patchy staining in the epithelial surface layer (black arrow). (G) CK19 showed diffuse staining in the basal cells of subepithelial layer (black arrow). (H) p16 showed scattered positive cells staining in the 1/3 of subepithelial layer (black arrow).

incidence of non-HPV-mediated OPSCC was greater than that of OSCC in terms of both positive rates and H scores.

## Expression correlation of CK7, CK19 and p16 in OPSCC/OSCC

In OPSCC, the co-expression of CK7 with CK19 ($P < 0.001$), CK7 with p16 ($P = 0.002$), and CK19 with p16 ($P = 0.019$) showed similar results (Table 4) while only the expression of CK7 was shown to be consistent with CK19 in OSCC (Kappa = 0.410, $P = 0.005$) (Table 5).

## DISCUSSION

### Expression relationship of CK7, CK19 and p16 in the pericancerous normal epithelium of OPSCC

In this study, we observed the expression of CK7, CK19 and p16 in the pericancerous surface epithelium of OPSCC. CK7 was located in the surface layer of epithelium (Fig. 1F). CK19 detected in the subepithelium layer and basal cells (Fig. 1G). p16, primarily coincides with the CK19 location, mainly located in the lower 1/3 to the entire layer of the epithelium (Fig. 1H). *Lee, Lee & Cho (2017)* also observed that both the simultaneous expression of CK7 in the epithelial surface cells and CK19 in the basal cells of the cervical intraepithelial

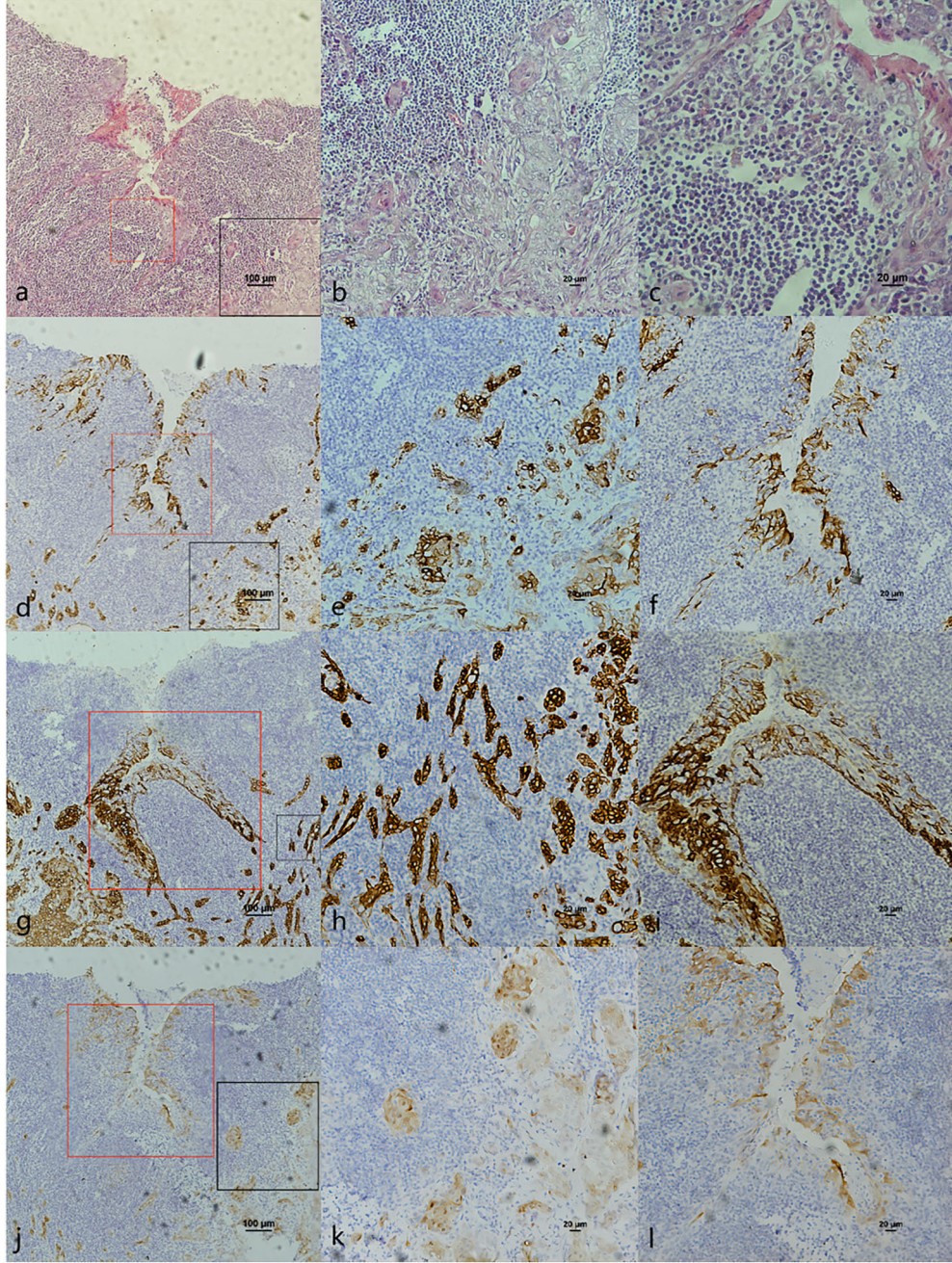

**Figure 3** CK7, CK19 and p16 staining in cancerous nest and pericancerous crypts normal epithelium of HPV-mediated (p16+) soft palate squamous cell carcinoma. (A) HE staining. The soft palatal mucosal epithelium falls inward and form crypt-like structure. (B) HE staining of cancerous nests staining (Fig. 4A black frame). (C) HE staining of crypt. Lymphatic follicular-like structures were visible below the soft palate crypt-like structure (Fig. 4A red frame). (D) CK7 expression in both cancerous nests and crypt epithelium. (E) CK7 showed patchy staining in cancerous nests (Fig. 4D black frame). (F) CK7 showed patchy staining in the crypt epithelium (Fig. 4D red frame). (G) CK19 expression in both cancerous nests and crypt epithelium. (H) CK19 showed diffuse staining in cancerous nests (Fig. 4G black frame). (I) CK19 showed diffuse staining in the crypt epithelium (Fig. 4G red frame). (J) p16 expression in both cancerous nests and crypt epithelium. (K) p16 showed patchy staining in cancerous nests (Fig. 4J black frame). (L) p16 showed patchy staining in the crypt epithelium (Fig. 4J red frame).

**Table 2 Statistical table of the expression combination of CK7, CK19 and p16 in OPSCC/OSCC.**

| Expression combination | | | OPSCC | | | OSCC | |
|---|---|---|---|---|---|---|---|
| CK7 | CK19 | p16 | Cancerous nests | Pericancerous surface normal epithelium | Pericancerous crypt normal epithelium | Cancerous nests | Pericancerous surface normal epithelium |
| + | + | + | 18 | 10 | 7 | 3 | 5 |
| + | + | − | 12 | 10 | 12 | 8 | 4 |
| − | + | + | 4 | 11 | 3 | 3 | 6 |
| + | − | + | 0 | 0 | 0 | 0 | 0 |
| + | − | − | 0 | 1 | 0 | 1 | 0 |
| − | + | − | 17 | 8 | 1 | 8 | 5 |
| − | − | + | 1 | 2 | 0 | 3 | 4 |
| − | − | − | 17 | 18 | 33 | 14 | 15 |

neoplasia (CIN). CK7 expression descends toward the basal layer (top-down expression), while CK19 ascends from the basal to superficial layer (bottom-up expression) as CIN progresses. Additionally, p16 expression coincides with CK19. They suggested that one potential explanation would be concurrent neoplastic transformation of CK7+ cells in the upper layer derived from SCJ and CK19+ basal stem cells in the lower layer infected by different or multiple high-risk HPV (HR-HPV).

However, *Woods et al. (2017)* found that CK7 was exclusively expressed in the surface layer of the crypt epithelium around the normal tonsil and the tonsil SCC, but CK7 did not appeared in the surface epithelium. Therefore, they believed that the carcinogenesis caused by HPV infection should originate from the crypt epithelium rather than the surface epithelium. In addition, in this study, CK7 was not only observed in the oropharyngeal crypt epithelium (Figs. 4F and 4F), but also in the surface epithelium (Fig. 1F). But further investigations are needed to investigate whether the expression of CK7 in the surface epithelium can serve as a new pathway for HPV infection.

It was reported that CK19 is expressed in the crypt epithelium of normal tonsil and the columnar cells of SCJ in the cervix (*Lee, Lee & Cho, 2017*; *Woods et al., 2017*). *In vitro* experiments, *Chatterjee et al. (2019)* discovered a significant upregulation of CK19 mRNA and protein in HPV-infected cervical and tonsillar cells, suggesting that these specific areas expressing CK19 are vulnerable to the development of cancer due to HPV infection. In this study, we also observed strong positive expression of CK19 in both the crypt epithelium (Fig. 4I) and surface epithelium (Fig. 1G) of OPSCC. We propose that CK19 could be served as an additional indicator for oropharyngeal crypt epithelial cells.

In addition, a study examining 262 cases of tonsil tissue without tumors showed that 25% of the cases had p16+ in the crypt epithelium while all the cases had p16− in the superficial epithelium (*Klingenberg et al., 2010*). However, we observed that p16 was positive in both the pericancerous crypt epithelium (Fig. 4L) and surface epithelium (Fig. 1D) of tonsil SCC. Furthermore, the presence of CK7 positive in surface epithelium observed above, indicates that the surface epithelium could potentially be acted as a

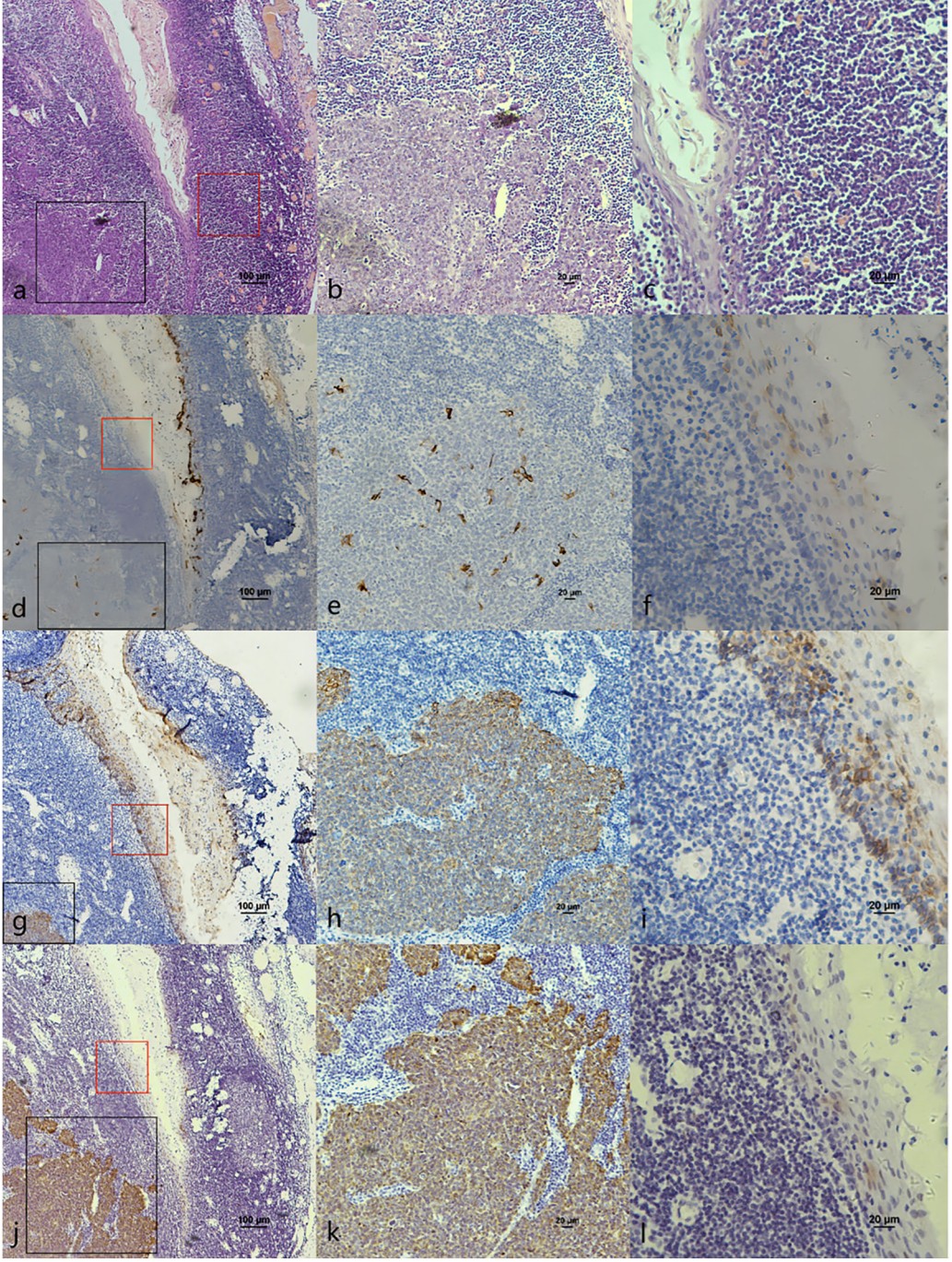

**Figure 4  CK7, CK19 and p16 staining in pericancerous crypts normal epithelium and cancerous nest of HPV-mediated (p16+) tonsil squamous cell carcinoma.** (A) HE staining. (B) HE staining of cancerous nests staining (Fig. 3A black frame). (C) HE staining of crypt. Crowded lymphocytes infiltration occurred below the crypt epithelium (Fig. 3A red frame). (D) CK7 expression in both cancerous nests and crypt epithelium. (E) CK7 showed scattered positive in cancerous nests (Fig. 3D black frame). (F) CK7 staining showed scattered positive cells in the crypt epithelium (Fig. 3D red frame). (G) CK19 expression in both cancerous nests and crypt epithelium. (H) CK19 showed diffuse staining in the cancerous nests (Fig. 3G black frame). (I) CK19 showed diffuse staining in the crypt epithelium (Fig. 3G red frame). (J) p16 expression in both cancerous nests and crypt epithelium. (K) p16 showed diffuse staining in the cancerous nests (Fig. 3J black frame). (L) p16 staining showed scattered positive cells in the crypt epithelium (Fig. 3J red frame).               

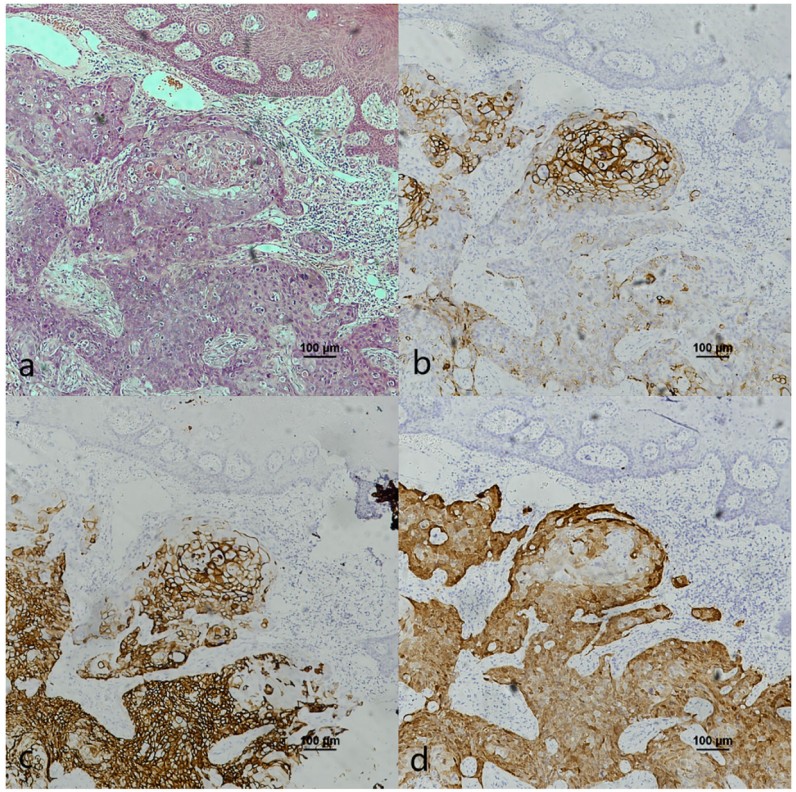

**Figure 5 CK7, CK19 and p16 staining in the cancerous nests of HPV- mediated (p16+) base of tongue squamous cell carcinoma.** (A) HE staining. (B) CK7 showed diffuse staining in cancerous nests. (C) CK19 showed diffuse staining in cancerous nests. (D) p16 showed diffuse staining in cancerous nests.

pathway for HPV infection. In the case of carcinogenesis caused by tonsils with HPV infection, the surface epithelium is infected in a reverse manner by HPV from cancerous nests, according with the traditional view that HR-HPV first infects exposed basal cells and transformed basal cells divide and migrate upward following the squamous epithelial differentiation program (*Lee, Lee & Cho, 2017*). Another possible explanation is that the presence of CK19 and p16 in the cancerous nests may inversely cause the surface epithelium to express CK19 and p16.

In this study, we noticed the presence of CK7+CK19+p16+ in the pericanerous crypt epithelium of tonsil and base of the tongue SCC in serial sections. Additionally, we discovered lymphoid follicles in the soft palate, together with the presence of crypt epithelial structure of CK7+CK19+p16+ deep into the follicle (Fig. 6). It was the first time that lymphoid follicles and crypt structure were observed in human being soft palate tissue. According to reports, lymphoid follicles have been observed in the soft palate of pig and cattle, but not in humans (*Casteleyn et al., 2011*). Observing the lymphoid follicles and crypt structure in the soft palate is challenging in HE sections because of their atypical nature compared to the structure in tonsil and base of the tongue. The structure may potentially be served as an anatomical basis for HPV-mediated carcinogenesis in the soft palate.
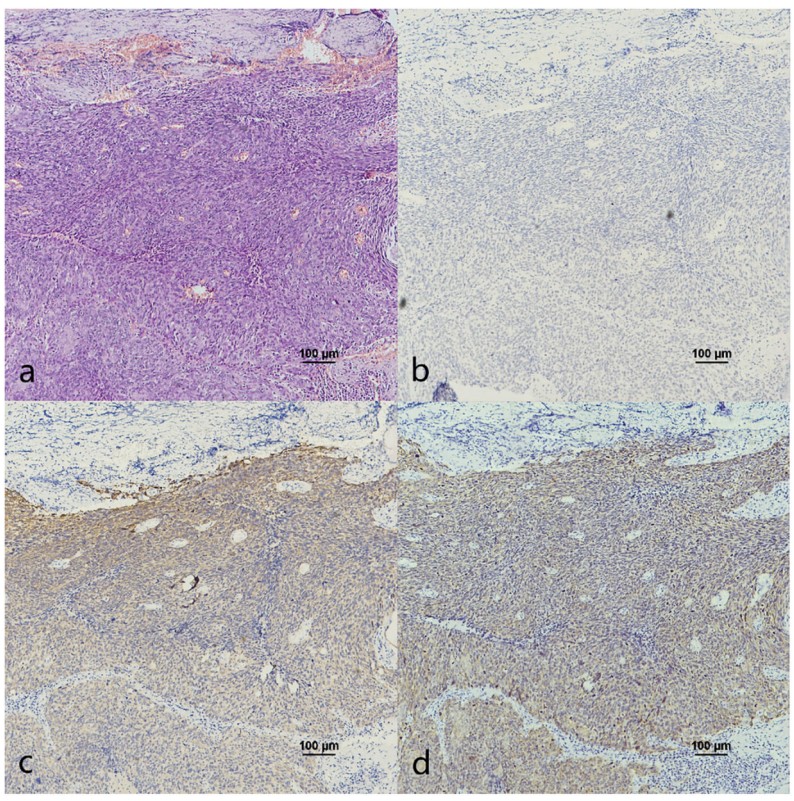

**Figure 6 CK7, CK19 and p16 in the cancerous nests of HPV-mediated (p16+) tonsil squamous cell carcinoma.** (A) HE staining. (B) CK7 expression in cancerous nests was negative. (C) CK19 showed diffuse staining in cancerous nests. (D) p16 showed diffuse staining in cancerous nests.

Based on the above-mentioned, we propose that the susceptibility of the oropharynx to HPV infection is caused by the specific expression of CK7 in the reticular epithelium of the oropharynx. The crypt epithelium of tonsil and base of the tongue near the lymphoid follicles is similar to the SCJ of the cervix and anus. In terms of anatomical structure, the crypt structure increases the chance of HPV remaining in the epithelium. The susceptibility of the crypt structure to HPV-mediated carcinogenesis is enhanced by the unique binding mechanism of CK7 and E7 (*Kanduc, 2002*; *Favia et al., 2004*; *Lucchese et al., 2010*).

## The relationship between OPSCC and expression of CK7, CK19 and p16

E7 is the predominant oncoprotein of HPV. HPV mRNA binds with the 6-mer peptide SEQIKA, which corresponds to amino acids 91–96 of the CK7 sequence. This interaction prevents the degradation of E7 mRNA. Only CK7 and E7 exhibit this specific response in the keratin family (*Kanduc, 2002*). CK7 is specifically expressed at the junction site in epithelium, such as the cervical SCJ epithelium and tonsil crypt epithelium, and it binds to E7 mRNA and locks its transcription at the junction site infected with HPV (*Mirkovic et al., 2015*; *Yang et al., 2015*; *Woods et al., 2017*). In contrast, CK19, as a heterodimer of

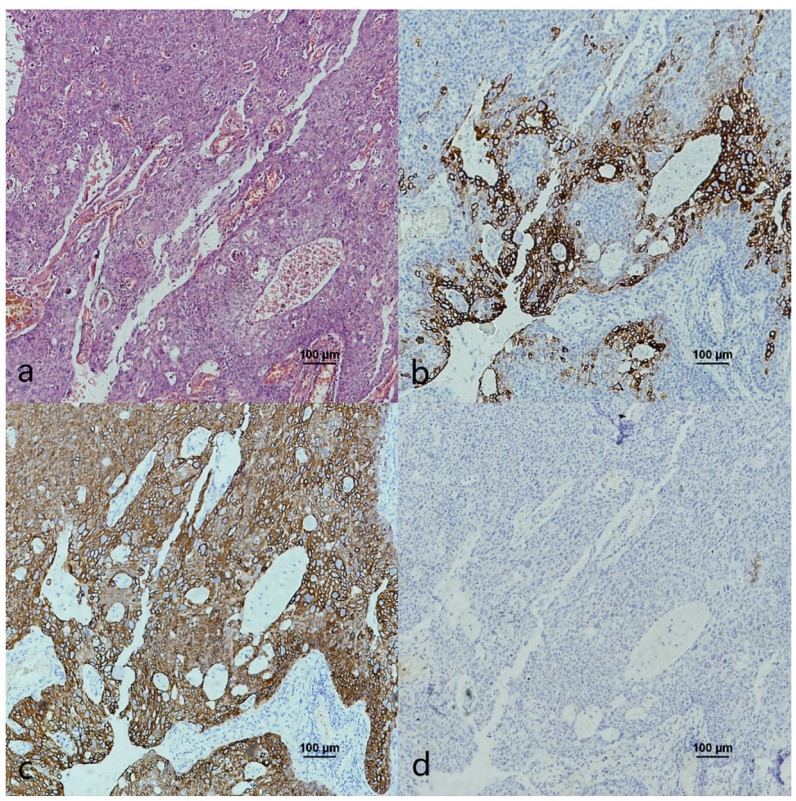

**Figure 7 CK7, CK19 and p16 staining in the cancerous nests of non-HPV-mediated soft palate squamous cell carcinoma.** (A) HE staining. (B) CK7 showed patchy staining in cancerous nests. (C) CK19 showed diffuse staining in cancerous nests. (D) p16 expression in cancerous nests was negative.

CK7, unlocks and promotes the translation of E7 mRNA into the oncoprotein by binding to the CK7 SEQIKA peptide (*Favia et al., 2004*; *Lucchese et al., 2010*). The E7 oncoprotein interacts with and promotes the degradation of pRb, relieving the inhibition of p16 transcription and causing p16 overexpression (*Chen et al., 2012*; *Serra & Chetty, 2018*). The mechanisms that CK7 and CK19 regulate E7 until p16 overexpression during HPV carcinogenesis are yet unknown. In this study, except CK7+CK19+p16+, the expression patterns of CK7−CK19+p16+ and CK7+CK19+p16− could also be found in both the cancerous nests and pericancerous normal epithelium of OPSCC. Surprisingly, only the expression pattern of CK7+CK19-p16+ could not be observed in any of the cases (Table 2 and File S1). The statistical results showed that expression level (positive rate and H-score) of CK7 and CK19 in HPV-mediated (p16+) OPSCC was higher than those in non-HPV-mediated OPSCC and OSCC (Table 3), and the pairwise expressions of the three proteins were all consistent in OPSCC (Table 4). This indicates a strong correlation of CK7, CK19 and p16 with the development of HPV-mediated OPSCC. Thus, we hypothesize the existence of a link termed the "CK7→CK19→p16 expression axis" during the process of CK7 and CK19 involved in the regulation of E7. However, due to the limited of study, it has not been verified whether this expression axis follows a strict temporal order.

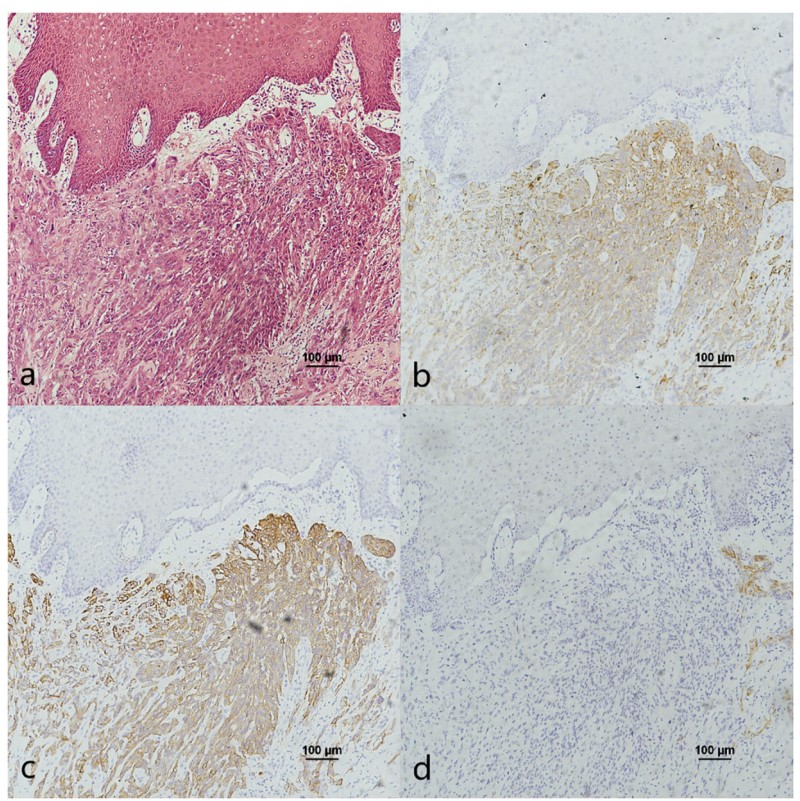

**Figure 8 CK7, CK19 and p16 staining in the cancerous nests of OSCC.** (A) HE staining. (B) CK7 showed diffuse staining in cancerous nests. (C) CK19 showed diffuse staining in cancerous nests. (D) Expression of p16 was negative in cancerous nests and scattered positive only in the tumour intercellular substance.

**Table 3 *P*-values for group differences in positive rate and H scores of CK7 /CK19 of HPV-mediated (p16+) OPSCC, non-HPV-mediated OPSCC and OSCC.**

|  | CK7 positive rate | CK7 H-score | CK19 positive rate | CK19 H-score |
|---|---|---|---|---|
| HPV mediated (p16+) OPSCC | 78.9% (15/19) | 80.3 ± 75.9 | 94.7% (18/19) | 160.3 ± 71.4 |
| Non-HPV-mediated OPSCC | 30.0% (15/50) | 20.5 ± 46.5 | 68.0% (34/50) | 78.3 ± 84.4 |
| OSCC | 32.5% (13/40) | 15.4 ± 38.0 | 52.5% (21/40) | 40.3 ± 66.5 |
| HPV mediated (p16+) OPSCC and non-HPV-mediated OPSCC | $P < 0.001$ | $P < 0.001$ | $P = 0.027$ | $P < 0.001$ |
| HPV mediated (p16+) OPSCC and OSCC | $P = 0.001$ |  | $P = 0.001$ |  |
| Non-HPV-mediated OPSCC and OSCC | $P = 0.799$ |  | $P = 0.001$ |  |

**Table 4 Values of Kappa and P of CK7, CK19 and p16 in OPSCC.**

| CK7 | CK19 + | CK19 − | Kappa | *P* | CK7 | p16 + | p16 − | Kappa | *P* | CK19 | p16 + | p16 − | Kappa | *P* |
|---|---|---|---|---|---|---|---|---|---|---|---|---|---|---|
| + | 30 | 0 | 0.4 | <0.001 | + | 18 | 11 | 0.3 | 0.002 | + | 26 | 26 | 0.22 | 0.019 |
| − | 21 | 18 | 27 |  | − | 10 | 30 | 72 |  | − | 3 | 14 | 2 |  |

Table 5 Values of Kappa and P of CK7, CK19 and p16 in OSCC.

| CK7 | CK19 | | Kappa | P | CK7 | p16 | | Kappa | P | CK19 | p16 | | Kappa | P |
|---|---|---|---|---|---|---|---|---|---|---|---|---|---|---|
| | + | − | | | | + | − | | | | + | − | | |
| + | 11 | 2 | 0.4 | 0.005 | + | 3 | 9 | 0.0 | 0.804 | + | 6 | 15 | 0.1 | 0.379 |
| − | 10 | 7 | 10 | | − | 6 | 22 | 38 | | − | 3 | 15 | 14 | |

The expression pattern of CK7+CK19+p16− was observed in both OPSCC (Fig. 7) and OSCC (Fig. 8). Probably because CK7 and CK19 have paired properties, they can be expressed simultaneously. And p16− indicates that the case might not be attributed to carcinogenesis resulting from HPV infection. We also observed the expression pattern of CK7−CK19+p16+ in the HPV-mediated (p16+) OPSCC (Fig. 6). A study on the carcinogenesis process of cervical cancer also found that CK7 was strongly expressed in the precancerous lesion, and with the cancerous progression, nearly 1/3 of the cases were CK7− but still CK19+ and along with p16+ (*Lee, Lee & Cho, 2017*).

There are several possibilities for this phenomenon. *Huang et al. (2016)* found that CK7 is a marker of transformation to malignancy in low-grade CIN, while p16 is not involved. Moreover, *Hashiguchi et al. (2019)* also reported that CK7 disappearing indicates the carcinogensis progression and poor prognosis in cervical cancer. A study showed that CK7 was expressed strongly and specifically in the tonsil crypt epithelium, while CK7− and p16+ were found in tonsil carcinoma (*Woods et al., 2017*). This study suggested that CK7 gradually disappears with tumor progression and cell differentiation. However, *Santoro et al. (2015)* provided the first evidence of a strong association between CK19 up-regulation and HR-HPV+OPSCC. They suggested that the CK19 test may be valuable in detecting HR-HPV infection. Based on the explanations provided above and our research findings, CK7 may be served as an "early event" and a necessary condition for HPV infection. As carcinogenesis progresses, the expression of CK7 gradually decreases. CK19 may be served as an intermediary between the preceding and the following event. And p16, closely related to CK19, is a "late event" in the carcinogenic process of HPV infection.

According to *Woods et al.*'s *(2017)* study, there is a strong correlation between CK7 and p16 in OPSCC. This finding aligns with our own data (Table 4). In addition, our investigation revealed a strong positive expression of CK7, CK19 and p16 in OPSCC (Figs. 3–5). This strong positive expression might represent the highest level of expression within the tumor, and then CK7 gradually vanished. Thus, we deducet that CK7 needs the participation of CK19 to cause HPV-mediated carcinogenesis. Without the facilitation of CK19 on E7mRNA translation, CK7 would be unable to induce p16 expression. Therefore, CK7 and CK19 have a significant impact in carcinogenesis caused by HPV.

In this study, we found that the CK7 expression level of non-HPV association OPSCC and OSCC was similar (Table 3). *Woods et al. (2022)* also found a similar situation in the comparison of HPV negative OPSCC and OSCC, suggesting that HPV negative OPSCC and OSCC may originate from surface epithelial cells with no or low expression of CK7. There is a potential alternative explanation that the carcinogenesis of non-HPV-mediated

OPSCC and OSCC is derived from the CK7+ salivary gland duct and duct epithelial cells, which may arise carcinogenesis even without HPV infection. Our previous study observed such SCJ structures in the vallate papilla of the base of the tongue (*Chen et al., 2022*). *Mestre et al. (2020)* also observed a structure similar to SCJ at the duct of the Von Ebner's gland of HPV-16 transgenic mice. Therefore, this form of carcinogenesis might originate from the SCJ-like structures such as the Von Ebner's gland at base of the tongue.

An interesting finding was that the CK19 expression level in OPSCC in the present study was higher in the non-HPV-mediated than that in OSCC (Table 3), similar to *Woods et al.*'s *(2022)* results. CK19 is normally expressed in the basal cells of nonkeratinized stratified squamous epithelium (*Barrett et al., 2005*; *Moll, Divo & Langbein, 2008*). We believe that the CK19 negative OSCC was derived from the keratinized squamous epithelium of the oral mucosa. This keratinized squamous epithelium usually lacks the expression of CK19, while the epithelium of the oropharynx is mostly nonkeratized stratified squamous epithelium with high expression of CK19. This result also further demonstrates the significance of the interaction between CK7 and CK19 in the development of HPV-mediated (p16+) OPSCC.

The findings of our study provided the possibility of "CK7→CK19→p16 expression axis" during the carcinogenesis of HPV-mediated OPSCC. However, one limitation of this study is that it is only a serial section study in which IHC was conducted, but no cytological or animal models experiments were performed. Currently, only cervical carcinoma conduct the cytologic study of these three proteins (*Kanduc, 2002*; *Favia et al., 2004*; *Lucchese et al., 2010*), while no in-depth cytological studies and even animal models have been performed on oropharyngeal carcinoma. The other limitation is that the present study lacks molecular studies to identify HPV status like HPV E7-HPV-mRNA *in situ* hybridization and HPV-DNA chips technology. Further research in molecular and cytologic experiments as well as animal models are needed in the future.

## CONCLUSIONS

Through the observation of normal oropharyngeal epithelium as well as squamous cell carcinoma, we suggest that the expression of CK7, CK19 and p16 is closely related to the HPV-mediated OPSCC and there may be an expression sequence of CK7→CK19→p16. Further work would include more molecular study, cytological experiments and animal models to prove these mechanisms.

## ACKNOWLEDGEMENTS

The authors gratefully acknowledge all the study participants and study staff for their help and cooperation during this study.

### Funding

This work was supported by the National Natural Science Foundation of China (No. 81660181), Natural Science Foundation of Guangxi Province (No. 2016GXNSFDA380002)

of China and Oral Health Promotion and Stomatology Development of The Western Clinical Scientific Research Foundation (No. CSA-W2022-08) of Chinese Stomatological Association. The funders had no role in study design, data collection and analysis, decision to publish, or preparation of the manuscript.

## Grant Disclosures

The following grant information was disclosed by the authors:
National Natural Science Foundation of China: 81660181.
Natural Science Foundation of Guangxi Province: 2016GXNSFDA380002.
Oral Health Promotion and Stomatology Development of The Western Clinical Scientific Research Foundation of Chinese Stomatological Association: CSA-W2022-08.

## Competing Interests

The authors declare that they have no competing interests.

## Author Contributions

- Qizhang Yan conceived and designed the experiments, performed the experiments, analyzed the data, prepared figures and/or tables, authored or reviewed drafts of the article, and approved the final draft.
- Pengning Chen performed the experiments, prepared figures and/or tables, authored or reviewed drafts of the article, and approved the final draft.
- Xinyu Chen performed the experiments, prepared figures and/or tables, authored or reviewed drafts of the article, and approved the final draft.
- Guanxi Chen performed the experiments, prepared figures and/or tables, and approved the final draft.
- Lin Luo performed the experiments, prepared figures and/or tables, and approved the final draft.
- Ping Ruan conceived and designed the experiments, authored or reviewed drafts of the article, and approved the final draft.
- Dahai Yu conceived and designed the experiments, analyzed the data, authored or reviewed drafts of the article, and approved the final draft.
- Mengyu Zhou conceived and designed the experiments, analyzed the data, authored or reviewed drafts of the article, and approved the final draft.
- Xiaojuan Zeng conceived and designed the experiments, analyzed the data, authored or reviewed drafts of the article, and approved the final draft.

## Human Ethics

The following information was supplied relating to ethical approvals (*i.e.*, approving body and any reference numbers):

Permission was obtained from the Ethics Committee of The First Affiliated Hospital of Guangxi Medical University (No. 2022021) and Guangxi Medical University College of Stomatology [No. 2022-KY-E- (149)].
## Data Availability

The raw measurements are available in the Supplemental File 1: Raw Data.

## Supplemental Information

Supplemental information for this article can be found online at http://dx.doi.org/10.7717/peerj.18286#supplemental-information.

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
