# Peer review of "Expression of CK7, CK19 and p16 in HPV-mediated oropharyngeal squamous cell carcinoma"

_PeerJ, doi:10.7717/peerj.18286_

## Round 0.1 · original submission · Major Revisions

· Academic Editor

Major Revisions

The reviewers found your manuscript interesting, however they had a number of significant concerns that need to be addressed. The main issue the reviewers identified was overstating your conclusions, specifically that CK7, CK19 and p16 are involved in the carcinogenesis and development of HPV-mediated OPSCC. This statement implies causality, which as the reviewers point out, cannot be concluded from your data. They suggest either revising your conclusions or performing additional studies such as western blotting, qPCR and other experiments evaluating the direct contribution of these proteins to HPV-mediated OPSCC. Additionally, the reviewers suggested that rather than using p16 as a proxy for HPV infection, detection of the E7 mRNA be used, since it is the gold standard. They requested that you explain the rationale for using OSCC as controls and that you include patients from different stages of the disease and perform analysis of the expression levels using stratified subgroups of patients. Lastly, the reviewers requested that you explain why you used the H score for your analysis, and commented on whether that was the most appropriate method, rather than a one-way ANOVA or Kruskal-Wallis.

Please, submit a detailed rebuttal which shows where and how you have taken all comments and suggestions into consideration. If you do not agree with some of the reviewers’ comments or suggestions, please explain why. Your rebuttal will be critical in making a final decision on your manuscript. Please, note also that your revised version may enter a new round of review by the same or by different reviewers. Therefore, I cannot guarantee that your manuscript will eventually be accepted.

**Language Note:** The review process has identified that the English language must be improved. PeerJ can provide language editing services - please contact us at [email protected] for pricing (be sure to provide your manuscript number and title). Alternatively, you should make your own arrangements to improve the language quality and provide details in your response letter. – PeerJ Staff

Reviewer 1 ·

Basic reporting

There are too many grammatical, spelling and formatting errors detected throughout the entire report. The language used was simple and understandable, albeit in some sentences it did not reflect academic writing style. In terms of formatting, the in-text citations were also wrongly written.

Experimental design

Not difficult and reproducible. H score was used as a method of scoring - could have used a better explanation for using this type of scoring; e.g for semi-quantitative analysis. The author was comparing 3 groups- OPSCCp16+, OPSCCp16-, and OSCC. The chosen statistical analysis tool is questionable. One way ANOVA or Kruskal-Wallis could be a better choice?

Validity of the findings

The conclusions were well-stated. However, the major limitation/ recommendation of this study was not discussed. p16 was mentioned as a surrogate marker for HPV infection, but utilisation of molecular studies are crucial to further validate the findings.

·

Basic reporting

In this report, the authors evaluated the expression of CK7, CK19 and p16 in tumor tissue sections from 19 HPV-positive OPSCC patients, 50 HPV-negative OPSCC patients and 40 OSCC patients, and observed the CK7+ CK19+ p16+, CK7-CK19+p16+ and CK7+CK19+p16- patterns but no CK7+ CK19-p16+ patterns in OPSCC patients. The authors concluded that CK7, CK19 and p16 are involved in the carcinogenesis and development of HPV-mediated OPSCC. The research purpose is meaningful. However, there are some critical issues which make the manuscript may not be qualified for PeerJ.

The first issue is over-concluding. The authors concluded that CK7, CK19 and p16 are involved in the carcinogenesis and development of HPV-mediated OPSCC. In fact, this study only studied the expressions of CK7, CK19 and p16, and expression data standalone does not qualify them as the contributors. To claim this conclusion, the manuscript needs to include studies evaluating the direction contribution or mechanism of CK7, CK19 or p16 to the development of HPV-mediated OPSCC. Given there are previous publications documenting the role of CK7, CK19 or p16 in OPSCC pathogenesis, the authors can only propose their data support CK7, CK19 or p16 might play roles in OPSCC pathogenesis.

The 2nd issue is study design. The authors want to speculate the expression order of CK7, CK19 and p16 during carcinogenesis. For this purpose, besides analyzing the cancerous and peri-cancerous tissues, patients of different stages, especially early stages should be included in this study and the expression correlation analysis should be done in stratified subgroups.

Experimental design

The authors want to speculate the expression order of CK7, CK19 and p16 during carcinogenesis. For this purpose, besides analyzing the cancerous and peri-cancerous tissues, patients of different stages, especially early stages should be included in this study and the expression correlation analysis should be done in stratified subgroups.

OSCC patients’ samples were used as controls, but the authors did not explain the rationale and did not reach a clear understanding about the differences between the two groups of patients. Recommend the authors to provide a clearer description of the comparisons between HPV-positive OPSCC and HPV-positive OSCC, as well as comparisons between HPV-negative OPSCC and HPV-negative OSCC.

Validity of the findings

The research data are well represented. The main issue is over concluding. The conclusions made could not be sufficiently supported by the presented data in the manuscript and should be revised.

The authors concluded that CK7, CK19 and p16 are involved in the carcinogenesis and development of HPV-mediated OPSCC. In fact, this study only studied the expressions of CK7, CK19 and p16, and expression data standalone does not qualify them as the contributors. To claim this conclusion, the manuscript needs to include studies evaluating the direction contribution or mechanism of CK7, CK19 or p16 to the development of HPV-mediated OPSCC. Given there are previous publications documenting the role of CK7, CK19 or p16 in OPSCC pathogenesis, the authors can only propose their data support CK7, CK19 or p16 might play roles in OPSCC pathogenesis.

Additional comments

Minor issues:
Scientific language needs to be significantly improved. Below are a few examples:
Line 96, "expresses" should be corrected to "is expressed."
Line 118, “CK7, CK19 and p16 were detected in the serial paraffin sections...”
Line 206, "CK7, CK19 or p16 was positively separated" is hard to understand. Should be rephrased.
Line 218, "expression consistence" is not a correct expression. "Expression correlation" may be more appropriate.

Reviewer 3 ·

Basic reporting

This manuscript investigates the localization of CK7, CK19 and p16 and the relationship among the three proteins in OPSCC by immunohistochemistry. The author further proposes the potential reason for the vulnerability of the oropharynx to HPV and CK7->CK19->p16 expression axis. The results and data analysis are straightforward, and the discussion is thorough. However, there are missing citations in introduction. There are no citations for line 69-71 “Since 1970… European and American countries. “and line 75 “In the United States, HPV infection was accounted for 45%-90% of OPSCC.”

Experimental design

However, only using one experimental method to study the three proteins is my major concern. Only one research method could potentially bring some bias. If possible, WB and qPCR can help quantifying the protein level and exploring the relationships better.

Validity of the findings

1. Abstract can be more precise. In the Methods, it is recommended to mention OSCC as controls, otherwise, OSCC just comes out of nowhere. In the Conclusions, “The CK7, CK19 and p16 are involved in the carcinogenesis and development of HPV-mediated OPSCC.” cannot fully reflect the results and the ideas of this manuscript. I feel the most important take home message is about the localization and the relationship in HPV-mediated OPSCC.
2. Missing citations in introduction. There are no citations for line 69-71 “Since 1970… European and American countries. “and line 75 “In the United States, HPV infection was accounted for 45%-90% of OPSCC.”
3. Line 195-197, the description seems not consistent with figure 3 legend. In the context, CK19 staining is shown in Figure 3c and p16 is shown in Figure 3d, while in the figure 3 legend, Figure 3g is CK19 and Figure 3j is p16.
4. Line 197-198, “the expression of CK7, CK19 and p16 in the pericancerous normal epithelium of OSCC was similar but not typical”. Do you have any potential explanations for this?
5. While the positive rate of CK7 and CK19 in HPV-mediated OPSCC is significantly higher, the negative condition still exists. How can this help the understanding of the roles of CK7 and CK19 in HPV-mediated OPSCC?
6. Line 283-295 focuses on the discussion of the relationship among E7, CK7 and CK19, and then transit to p16. While p16 help differentiating HPV+ and HPV- OPSCC, E7 mRNA is still the gold standard and relevant experiment about E7 mRNA level can better identify the role of CK7 and CK19.
7. I suggest to at least have some preliminary results to support “CK7->CK19->p16 expression axis”.

Additional comments

1. Line 171 “2 test”, is this a typo?
2. Table 1, I didn't see any numbers in "Age" row.
3. Format issues (100×) in several figure legends.

---

## Round 0.2 · Minor Revisions

· Academic Editor

Minor Revisions

Thank you for thoroughly addressing the reviewers’ comments. Overall, the reviewers found your revised manuscript much improved. One of the reviewers who had also reviewed the original version, had a couple of additional concerns. First, the reviewer suggested that you increase the sample size in an effort to strengthen your study. Additionally, the reviewer asked why you chose to perform an exclusively cytological study rather than a molecular study, so please address this concern in your rebuttal and revised manuscript

Please, submit a detailed rebuttal which shows where and how you have taken all comments and suggestions into consideration. If you do not agree with some of the reviewers’ comments or suggestions, please explain why. Your rebuttal will be critical in making a final decision on your manuscript. Please, note also that your revised version may enter a new round of review by the same or by different reviewers.

Reviewer 1 ·

Basic reporting

The study provides valuable insights into the expression patterns of CK7, CK19, and p16 in HPV-mediated OPSCC. There are still grammatical errors present in the manuscript. For example under statistical analysis -contingency tables in analyze should be changed to 'to analyze', under discussion point no. 1 - with CIN progresses should be 'as CIN progresses'. Please proofread the article before submitting.

Experimental design

The experimental design is generally sound, but could be strengthened by increasing the sample size.

Validity of the findings

No comment

Additional comments

The conclusion/recommendation is rather weak. Why do a cytological study? It doesn't give a more specific result. To reiterate about molecular study makes much more sense

·

Basic reporting

My previous comments were fully addressed. No further comments.

Experimental design

No further comments.

Validity of the findings

My previous comments were fully addressed. No further comments.

Additional comments

My previous comments were fully addressed. No further comments.

Reviewer 3 ·

Basic reporting

Thanks for the thorough work and for addressing my comments. The revisions have successfully addressed the concerns raised, and the manuscript is now suitable for publication.

Experimental design

The revisions have successfully addressed the concerns raised, and the manuscript is now suitable for publication.

Validity of the findings

The revisions have successfully addressed the concerns raised, and the manuscript is now suitable for publication.

---

## Round 0.3 · accepted · Accept

· Academic Editor

Accept

Thank you for satisfactorily addressing the reviewer's comments and greatly improving your manuscript.

Reviewer 1 ·

Basic reporting

Corrections from the previous review have been addressed

Experimental design

No commentDat

Validity of the findings

Data were provided. All has been mentioned and addressed in the previous review.

Additional comments

No comment.